Evaluation of droplet deposition parameters based on the Genetic-Otsu algorithm

Meng Yanhua 1
Liu Xinchao 2
Chen Wei 1
Du Xintao 1
Zhang Yifan 1
Sun Rui 3
Han Yuxing 4 yuxinghan@sz.tsinghua.edu.cn
1 School of Mechanical Engineering, Anyang Institute of Technology , Anyang, Henan Province , China
2 School of Electronic Engineering, South China Agricultural University , Guangzhou , China
3 China Agro-technological Extension Association , Beijing , China
4 Tsinghua Shenzhen International Graduate School , Shenzhen , China
Kutlu Imren
Electronic publication date: 2024 Sep 18
Publication date: 2024
Volume: 12
Electronic Location ID: e18036
Received 2024 Apr 16; Accepted 2024 Aug 12
Copyright: © 2024 Meng et al.
Copyright year: 2024
Copyright holder: Meng et al.
License: This is an open access article distributed under the terms of the Creative Commons Attribution License, which permits unrestricted use, distribution, reproduction and adaptation in any medium and for any purpose provided that it is properly attributed. For attribution, the original author(s), title, publication source (PeerJ) and either DOI or URL of the article must be cited.
License URL: https://creativecommons.org/licenses/by/4.0/

Keywords: UAV, Droplet extraction, WSP, Deposition parameters, Genetic algorithm, Otsu threshold method

Funding: National Natural Science Foundation of China 32201659 Shenzhen Startup Funding QD2023014C This work was supported by the National Natural Science Foundation of China (No. 32201659) and Shenzhen Startup Funding (No. QD2023014C). The funders had no role in study design, data collection and analysis, decision to publish, or preparation of the manuscript.

==============================
Pesticide spraying is a cost-effective way to control crop pests and diseases. The effectiveness of this method relies on the deposition and distribution of the spray droplets within the targeted application area. There is a critical need for an accurate and stable detection algorithm to evaluate the liquid droplet deposition parameters on the water-sensitive paper (WSP) and reduce the impact of image noise. This study acquired 90 WSP samples with diverse coverage through field spraying experiments. The droplets on the WSP were subsequently isolated, and the coverage and density were computed, employing the fixed threshold method, the Otsu threshold method, and our Genetic-Otsu threshold method. Based on the benchmark of manually measured data, an error analysis was conducted on the accuracy of three methods, and a comprehensive evaluation was carried out. The relative error results indicate that the Genetic-Otsu method proposed in this research demonstrates superior performance in detecting droplet coverage and density. The relative errors of droplet density in the sparse, medium, and dense droplet groups are 2.7%, 1.5%, and 2.0%, respectively. The relative errors of droplet coverage are 1.5%, 0.88%, and 1.2%, respectively. These results demonstrate that the Genetic-Otsu algorithm outperforms the other two algorithms. The proposed algorithm effectively identifies small-sized droplets and accurately distinguishes the multiple independent contours of adjacent droplets even in dense droplet groups, demonstrating excellent performance. Overall, the Genetic-Otsu algorithm offered a reliable solution for detecting droplet deposition parameters on WSP, providing an efficient tool for evaluating droplet deposition parameters in UAV pesticide spraying applications.

Introduction

The chemical control method is one of the crucial means to manage crop diseases and pests, ensuring food security (Ratnadass et al., 2011). According to data from the Food and Agriculture Organization of the United Nations (FAO) (FAOSTAT, 2022), approximately 3,319,054 tons of pesticides were used globally on average each year over the past decade, with China alone using around 302,830.8 tons of insecticides annually, accounting for 9.12% of the world’s annual average pesticide usage.

Unmanned aerial vehicles (UAVs), ground-based equipment, and manual spraying devices are the primary tools for pesticide application, and their effectiveness is crucial for successful pest and disease control (Chen et al., 2021; Viret et al., 2003). Research indicates that the deposition distribution of spray droplets on the target is a key indicator for evaluating the quality of spraying (Shan et al., 2021; Zheng et al., 2021). Assessing the droplet deposition distribution is of significant importance in enhancing pesticide utilization efficiency and ensuring effective control of plant diseases and pests (Hou et al., 2019; Soheilifard et al., 2020).

Currently, there are two main methods for detecting the parameters of droplet deposition: direct detection method and indirect detection method (Srinivasarao et al., 2021; Zheng et al., 2017). The direct detection method usually employs tracer substances, instead of pesticides, for spraying on crops. Subsequently, the tracer substances are washed off from the plants or leaves using distilled water or organic reagents, and the droplet deposition amount is determined through spectroscopic analysis and other techniques (Palma, Cunha & de Santana, 2023; Wang et al., 2021; Yuan et al., 2012). Nevertheless, the direct detection method is characterized by intricate operational procedures, and high costs, and is not conducive to rapid field testing (Wang et al., 2019). The indirect detection method refers to the use of artificial media such as water-sensitive paper (WSP) and copperplate paper instead of leaves, which are placed in the spray area to collect droplets. By analyzing the deposition parameters of droplets on the sampling media, the deposition status of droplets in the spray area can be indirectly assessed (Sies et al., 2017; Yang et al., 2022). Compared to direct detection methods, utilizing media such as WSP for detecting droplet deposition characteristics is more convenient, cost-effective, and provides superior detection results (Salyani et al., 2013; Wang et al., 2019). Therefore, it is widely employed in the assessment of droplet deposition characteristics.

To rapidly and accurately detect the droplet deposition parameters on WSP, some scholars have proposed image processing-based methods for assessing droplet parameters on WSP. In 2016, Ferguson et al. (2016) designed a WSP droplet analysis application named SnapCard for smartphones. Through a comparison with five other image-processing software tools, they showed that SnapCard could efficiently quantify droplet coverage without requiring costly software or intricate laboratory procedures (Ferguson et al., 2016). In 2020, Özlüoymak & Bolat (2020) introduced an image processing software integrated with a WSP conveyor belt system. Through adjustments in nozzle types, spraying agents, and conveyor belt speeds to replicate field spraying scenarios, they determined the spray deposition area. This software facilitated rapid evaluations of spray coverage and droplet quantities (Özlüoymak & Bolat, 2020). In 2021, Brandoli et al. (2021) presented a new image analysis software named Dropleaf, which indirectly assessed pesticide coverage by analyzing droplet areas on WSP. However, these methods use fixed or automatic threshold methods in the binarization process. Their capabilities in extracting small-sized droplets are limited, and they are susceptible to noise interference. This poses a challenge to the accurate assessment of droplet deposition parameters.

The Otsu method is an efficient and simple image segmentation algorithm widely used in various image processing tasks (Xu et al., 2011; Yousefi, 2015). However, when processing complex images, the Otsu method may be disturbed by noise and is not effective for droplet segmentation in complex environments. As an optimization method grounded in natural selection and genetic mechanisms, the genetic algorithm exhibits robust anti-noise capabilities and high adaptability. It is extensively utilized in the research of optimization algorithm performance (Binh, Loi & Thuy, 2012; Maulik, 2009).

To enhance the accuracy of detecting droplet deposition parameters and minimize the influence of image noise on their precision, this study suggests a droplet extraction method with superior stability and effective segmentation performance. By combining the genetic algorithm and Otsu thresholding method, our approach substantially mitigates the impact of noise on the algorithm, facilitating the efficient extraction of droplets on WSP and leading to precise evaluation of droplet deposition parameters. Through a comparative analysis of the performance of commonly used droplet extraction algorithms with the proposed algorithm on WSP containing three distinct levels of droplet deposition, the effectiveness of the algorithm presented in this study is further validated.

Materials and Methods

Spraying platform

The UAV employed in this study is the 3WQFTX-101S intelligent electric multi-rotor plant protection UAV manufactured by Henan Anyang Quanfeng Aviation Plant Protection Technology Co., Ltd., utilized as the spraying platform (Fig. 1). Table 1 shows the key performance parameters of this multi-rotor plant protection UAV, featuring a spraying system comprising four XR TEEJET 110015VS pressure nozzles with spray angles of 110°.

Figure 1 Quanfeng aviation 3WQFTX-101S intelligent electric multi-rotor plant protection UAV.

Table 1 Main performance parameters of the UAV.

Main parameters	Value	
Size (m)	1.37 * 1.37 * 0.65	
Spraying height (m)	1–3	
Spraying width (m)	3–5	
Volume of medicine box (L)	10	
Number of nozzles	4	
Weight (kg)	12 ± 1	
Maximum load (L)	10	
Spraying flow rate (L/min)	1.92–2.36	
Nozzle type	XR TEEJET 110015VS	

Experimental design

The experiment took place at the Anyang Institute of Technology in Anyang City, Henan Province. A vacant area measuring 30 m × 10 m was designated as the spraying zone. Throughout the experiments employing the spraying platform, the WSP (76 mm × 26 mm) was placed symmetrically along both sides of the UAV flight path, with five sampling points on each side, amounting to a total of 10 sampling points. The spacing between WSPs on one side in the vertical flight direction was 0.4 m. The flight path and sampling point configuration are depicted in Fig. 2. Three different flight speeds were utilized for the spraying experiments to gather sample data at various coverage. The spray test was repeated three times with the same WSP layout under the same speed conditions to ensure test accuracy. Table 2 presents the main parameter settings of the UAV platform during field application.

Figure 2 Field sampling point set up.

Table 2 The main parameter settings of the UAV platform during field application.

Height (m)	Speed (m/s)	Total nozzle flow (L/min)	Sample size	
2	2	2.36	30	
2	3	2.36	30	
2	4	2.36	30	

Sample collection and processing

After the UAV completed the spraying operation for 10 min, the WSP samples were sealed and dried using labeled kraft paper envelopes. Subsequently, all the WSPs were scanned using an HP SCANJET G4050 high-definition image scanner to obtain sample images at a resolution of 96 DPI. Finally, the collected 90 WSP images were categorized based on the droplet pixel area (PA) into dense droplets group (40,000 < PA < 90,000), medium droplets group (12,000 < PA ≤ 40,000), and sparse droplets group (PA ≤ 12,000). Figure 3 displays the WSP sample images of these three droplet groups, which will be used for the subsequent analysis of droplet coverage and density.

Figure 3 Example of the WSP image group.

Evaluation metrics for droplet deposition effectiveness

Droplet coverage and density are critical metrics for evaluating deposition uniformity and are essential for accurately assessing spray quality (Lv et al., 2019). Droplet coverage represents the ratio of the total area covered by all droplet particles deposited on the target surface to the total area of the target surface. The formula for evaluating droplet coverage is as follows:

(1) C=ASAP×100%

where C is the droplet coverage; AS is the total area of droplets; and Ap is the area of WSP.

Droplet density refers to the number of droplets deposited on a unit area of the target surface. The evaluation index formula is as follows:

(2) D=NA×100%

where D is the droplet density; N is the number of droplet particles on the target; and A is the area of WSP.

Fixed threshold segmentation methods

The fixed threshold method is a widely adopted image processing technique utilized in the fields of image segmentation, contour extraction, and feature extraction (Qi et al., 2022). This method operates by setting a fixed threshold and analyzing the gray-level distribution characteristics of the image. The algorithm scans each pixel value in the image, comparing it with the pre-defined threshold. Pixels exceeding the threshold are set to the maximum gray level value, while pixels below the threshold are set to the minimum gray level value, resulting in image binarization. The basic idea is as follows:

(3) G(x,y)={255,f(x,y)≥T0,f(x,y)<T

where T is the fixed threshold, f(x, y) is the pixel point gray value.

WPS is a highly sensitive professional test paper that exhibits a color reaction upon contact with water, creating a distinct color contrast between the paper and the droplets (Fox et al., 2003). Based on this color contrast, the threshold was set as a constant value of 0.60, corresponding to a value of 153 within the 8-bit range [0,255] (Brandoli et al., 2021).

Otsu threshold segmentation methods

The Otsu thresholding method, known as the maximum between-class variance method, is a widely employed image binarization segmentation algorithm (Xu et al., 2011). The core principle of the Otsu algorithm is to compute the intra-class variance and inter-class variance of the image at various thresholds to identify the optimal threshold that maximizes the inter-class variance. Initially, the algorithm classifies each pixel in the image based on its grayscale value. Subsequently, it calculates the probabilities of pixels being categorized as foreground and background separately, along with the cumulative mean of the target grayscale level. The algorithm then determines the optimal threshold value by maximizing the inter-class variance between the foreground and background. Assuming a threshold value of k, the expression for the cumulative mean m of the gray level K is:

(4) P1=∑i=0k⁡pi,P2=∑i=k+1255⁡pi,m=∑i=0k⁡ipi

where pi is the probability of the grayscale value and being i, P1 is the probability of a pixel being assigned to the background region, P2 is the probability of being assigned to the target region.

(5) σ2(k)=max0≤k≤255(mG∗P1−m)2P1∗P2

where mG represents the average grayscale value of the entire image, σ2 (k) is the maximum inter-class variance, and K is the optimal threshold value.

Threshold segmentation method based on the Genetic-Otsu algorithm

The genetic algorithm is a stochastic global search optimization method inspired by natural selection and genetic processes in biological systems (Mirjalili, 2019). Commencing with an initial population, the genetic algorithm utilizes a combination of random selection, crossover, and mutation operations to iteratively generate a population of individuals that exhibit improved adaptation to the prevailing environment. This evolutionary process enables the population to navigate towards more advantageous regions within the search space, gradually converging towards a subset of individuals that are optimally aligned with the environmental conditions.

The genetic algorithm is adept at globally searching for optimal solutions, irrespective of the mathematical characteristics of the problem. It is especially effective for tackling complex, multi-parameter, and nonlinear problems. The application of the Genetic-Otsu algorithm in extracting droplets from WSP is exemplified in Fig. 4.

Figure 4 Threshold segmentation method based on Genetic-OTSU algorithm.

Morphological processing

Morphological processing stands out as a highly utilized technique in the realm of image processing. Its basic idea is to use a special structuring element to measure or extract the corresponding shapes or features in the input image, to further analyze the image and recognize objects (Chanda, 2008). This method is commonly used in the preprocessing and postprocessing of images and is an effective image enhancement technique. It comprises essential operations such as erosion, dilation, opening, and closing.

Erosion operation is a process that “shrinks” the foreground area in an image, which can be used to eliminate edges and noise, making the image edges smoother. Dilation is a process that “expands” the foreground area in an image, making the image structure more complete. The opening operation begins with erosion followed by dilation, aiming to reduce noise and refine image edges. This process helps in smoothing irregularities and improving image quality. Conversely, the closing operation starts with dilation followed by erosion. This operation fills holes in the image and improves image segmentation. By closing small breaks in the image structure, this operation enhances the continuity and completeness of image features (Zhang, 2009).

In this study, the droplets on the WSP were extracted, and it was observed that there were many holes inside the droplet contours after threshold segmentation. This could affect the accuracy of droplet parameter calculation. To address this issue, a morphological closing operation was applied to optimize the segmented image, resulting in more complete and accurate droplet contours. Figure 5 compares the effects of droplet contours before and after threshold segmentation and morphological closing operation.

Figure 5 Comparison between before and after morphological closing treatment.

Data processing and analysis

In this study, Adobe Photoshop 2024 software was used to manually measure 90 WSP samples. For each sample, three researchers performed pixel-level contour extraction. By averaging the results of the three extractions, we calculated the actual values of droplet coverage and density.

Based on the manual measurement data, the accuracy error of the three methods is calculated. In order to fully manifest the disparities among the data, Tukey’s method was employed to assess the differences among treatments in a one-way analysis of variance, with a significance level set at 0.05.

Additionally, OriginLab 2021 software was employed to plot the data, thereby enhancing the visualization of data distribution and trends.

Results

Evaluation results of droplet coverage

Droplet coverage is one of the important indicators for evaluating the effectiveness of drone spraying. It reflects the uniformity and effectiveness of droplets on the target. Accurate detection of droplet coverage is crucial for optimizing drone parameters and improving pesticide utilization efficiency.

Figure 6 illustrates the relative errors in droplet coverage calculated by three threshold segmentation algorithms across three different droplet groups, compared to manual measurement results. The results indicate that, across the three groups, the Genetic-Otsu threshold method proposed in this study has lower segmentation errors and better extraction performance compared to the fixed threshold method and the Otsu threshold method. Specifically, in the sparse droplets group, the extraction performance of the three methods is comparable, and there is no significant difference in the relative error of droplet coverage results.

Figure 6 Results of droplet coverage for the three methods.

(A) Relative error of droplet coverage for the three methods on the few droplet groups. (B) Relative error of droplet coverage for the three methods on the medium droplet groups. (C) Relative error of droplet coverage for the three methods on the massive droplet groups. (D) Mean relative error of droplet coverage for the three methods on the three droplet groups. Different lowercase letters indicate significant differences among different treatments at the 0.05 level by Tukey’s test.

In the medium droplets group, the Genetic-Otsu threshold method proposed in this study yields significantly lower relative errors in droplet coverage compared to the fixed threshold method and the Otsu threshold method, with an average error of only 0.9% compared to manually calculated results. The average relative errors in droplet coverage obtained by the fixed threshold method and the Otsu threshold method do not show significant differences, with the Otsu threshold method slightly outperforming the fixed threshold method.

In the dense droplets group, the extraction performance of the fixed threshold method is poor, showing a significant difference in relative error compared to the Otsu threshold method and the Genetic-Otsu threshold method, with a relative error of 4.7%. Conversely, both the Genetic-Otsu threshold method and the conventional Otsu threshold method exhibit high segmentation accuracy in dense droplet groups, with no significant discernible difference between the two methods. The proposed algorithm in this study has a lower relative error compared to the Otsu threshold method, with only a 1.2% relative error.

Evaluation results of droplet density

Droplet density is an important evaluation indicator for the effectiveness of drone spraying and a key factor in determining the effectiveness of pesticide control. Figure 7 shows the relative errors in droplet density calculated by three threshold segmentation algorithms across three different droplet groups, compared to manual measurement results. The results indicate that across the three droplet groups, the Genetic-Otsu threshold method proposed in this study demonstrates outstanding extraction performance and outperforms both the fixed and traditional Otsu threshold methods in calculating droplet density.

Figure 7 Results of droplet density for the three methods.

(A) Relative error of droplet density for the three methods on the few droplet groups. (B) Relative error of droplet density for the three methods on the medium droplet groups. (C) Relative error of droplet density for the three methods on the massive droplet groups. (D) Mean relative error of droplet density for the three methods on the three droplet groups. Different lowercase letters indicate significant differences among different treatments at the 0.05 level by Tukey’s test.

In the sparse droplets group, the relative error of droplet density obtained using the fixed threshold method is significantly higher than that of the Otsu threshold method and the Genetic-Otsu threshold method, with an error of up to 19.7%. The results of the Otsu threshold method and the Genetic-Otsu threshold method are comparable, with no significant difference. In the medium and the dense droplet groups, there are significant differences in the relative errors of droplet density among the three methods. The Genetic-Otsu threshold method has the lowest average error in droplet density, followed by the Otsu threshold method, while the average relative error of the fixed threshold method is above 10%, indicating poorer performance.

Visualization of droplet extraction performance

Figure 8 illustrates the extraction results of three droplet extraction algorithms in three droplet groups. The results indicate that, across the three droplet groups, the fixed threshold method and the traditional Otsu threshold method both struggle to extract small-sized droplets. In contrast, across the three droplet groups, the Genetic-Otsu method proposed in this study shows effective extraction of small droplets and demonstrates superior extraction performance.

Figure 8 Small size droplet extraction results.

As illustrates in Fig. 9, with the increase in droplet density, both the fixed threshold method and the Otsu threshold method are prone to identifying the contours of multiple independent droplets that are adjacent but not in direct contact as a same droplet, leading to higher calculation errors in droplet deposition parameters. In contrast, the Genetic-Otsu threshold method proposed in this study exhibits excellent droplet extraction performance in all three droplet groups. This algorithm not only can effectively extract small-sized droplets but also can differentiate the multiple independent contours of adjacent droplets, thereby enabling accurate calculation of droplet deposition parameters.

Figure 9 Results of the three droplet extraction methods.

Discussion

The genetic algorithm is a global search algorithm that can extensively explore the solution space and avoid getting trapped in local optima. In this study, the problem of droplet extraction is regarded as an optimization problem, aiming to find a suitable segmentation threshold that can achieve the best segmentation effect for the image. Subsequently, the genetic algorithm is employed to search for this optimal threshold. By performing random selection, crossover, and mutation operations on candidate thresholds, the genetic algorithm continuously explores the solution space and eventually identifies the most suitable threshold. Due to its parallelism and robustness, the genetic algorithm can largely overcome some limitations of traditional algorithms and produce satisfactory results, especially in complex image scenarios or in the presence of significant image noise.

Moreover, once the algorithm extracts the outlines of droplets, the presence of glare phenomenon on the liquid droplets leads to the generation of numerous holes in the extracted droplet outlines, consequently diminishing the coverage of the liquid droplets. In this study, employing dilation followed by erosion operations enables the effective filling of small holes within the outlines, ensuring precise extraction of the liquid droplet outlines. This approach capitalizes on the attributes of morphological operations to adeptly tackle contour defects induced by glare, thereby enhancing the overall accuracy of liquid droplet outline extraction.

By utilizing a genetic algorithm to optimize the threshold selection process of the Otsu method, more accurate and stable results have been obtained at different coverage rates. Additionally, the use of morphological processing can better address the issue of extracting droplet contours in complex scenes. This comprehensive approach can enhance the overall algorithm performance, providing more precise and stable information on liquid droplet deposition parameters for subsequent applications.

Conclusion

To accurately reflect the distribution of droplet deposition in the sprayed area after drone spraying, this study developed a droplet deposition parameter evaluation method based on a genetic algorithm and Otsu thresholding method and conducted experimental verification. The main research findings are as follows: In the calculation of droplet coverage and density, the Genetic-Otsu thresholding method proposed in this study outperformed both the fixed thresholding method and the Otsu thresholding method across three droplet groups. Comparing with manually measured data, the average errors in droplet coverage and density were found to be 1.5% and 2.7% for the sparse droplets group, 0.88% and 1.8% for the medium droplets group, and 1.2% and 2.0% for the dense droplets group. These results demonstrate lower segmentation errors and improved extraction performance with the Genetic-Otsu method as compared to the other threshold methods.

The Genetic-Otsu thresholding method proposed in this study can effectively extract small-sized droplets, at high densities, and can accurately segment agglomerated droplets, significantly improving the algorithm’s accuracy.

In summary, the Genetic-Otsu thresholding method proposed in this study can effectively extract the droplet contours on WSP, achieve an accurate evaluation of droplet deposition parameters, and provide an efficient detection method for evaluating droplet deposition parameters.

Supplemental Information

Supplemental Information 1 Raw data.

Additional Information and Declarations

Competing Interests

Author Contributions

Data Availability

The authors declare that they have no competing interests.

Yanhua Meng conceived and designed the experiments, performed the experiments, authored or reviewed drafts of the article, and approved the final draft.

Xinchao Liu performed the experiments, analyzed the data, authored or reviewed drafts of the article, and approved the final draft.

Wei Chen analyzed the data, prepared figures and/or tables, authored or reviewed drafts of the article, and approved the final draft.

Xintao Du performed the experiments, analyzed the data, prepared figures and/or tables, and approved the final draft.

Yifan Zhang analyzed the data, prepared figures and/or tables, and approved the final draft.

Rui Sun conceived and designed the experiments, authored or reviewed drafts of the article, and approved the final draft.

Yuxing Han conceived and designed the experiments, authored or reviewed drafts of the article, and approved the final draft.

The following information was supplied regarding data availability:

The raw data is available in the Supplemental File.

The code is available in GitHub and Zenodo:

- https://github.com/Liuxinchao123/Code.git

- Liuxinchao123. (2024). Liuxinchao123/Code: Droplet extraction (v1.0.0). Zenodo. https://doi.org/10.5281/zenodo.13137654.

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
