# Peer review of "Evaluation of droplet deposition parameters based on the Genetic-Otsu algorithm"

_PeerJ, doi:10.7717/peerj.18036_

## Round 0.1 · original submission · Major Revisions

Make major revisions, taking into account reviewers' comments.

·

Basic reporting

The English language used in the paper is good and professional.
Literature references were sufficient for the context.
The article structure fits the Journal recommendation.

Experimental design

The methods and treatments used were not complex, so there was nothing to add.

Validity of the findings

The article brings simple research but adds important information related to a gap in the evaluation of pesticide applications.

Additional comments

Lines 19 - 21: I believe that this research is not only applicable to treatments performed by UAVs but also to any spraying methods for agricultural purposes. So I see no point in starting the summary with this information, which is related only to UAVs.
Line 29 "errors": There is a lack of a clear description of what the basic information was based on to compare with the three methods. And then, state that one was more accurate than the other.
Line 56-57: I would consider both methods more as complementary evaluations, not substitutable. On the one hand, deposits represent the amount of active ingredient that reaches the target, without information on the quality of distribution on the target surfaces. On the other hand, coverage, represents the distribution of droplets on the surfaces, considering the coverage and density of the droplets.
Line 100: The nozzle named XR1100115VS doesn't even exist... Was it a XR 11001VS or a XR 110015VS model? If it was the model XR11001VS being used, the only spray angle accepted is 110º.
Why was it mentioned the angle 80º?
Line 110: To achieve this total flow rate for the four nozzles (0.59 L/min for each one), I believe it was used the XR11001VS model, at the pressure of 100 psi.
Line 116: What was the resolution used for the WSP images (pixels per inch - ppi)?
Line 252: How is it possible to state that the method can differentiate droplets? Is there any image that can help to show this information?
A treated WSP image could help, showing the droplets deposited on the WSP and the contours made by the method, showing the ability to segregate droplets from a single spot with irregular contours, which refer to several agglutinated droplets.
Line 285 "average errors": What were these errors pointed out about? What was the basic/correct information based on to make the comparison possible?
Line 291 "accurately segment agglomerated droplets": For this statement, I would like to see some figures of the method acting on a WSP, showing the background with some spots of drops with complex morphology and the droplets identified by the method.

·

Basic reporting

The English language is mostly clear throughout the manuscript, with only some minor issues. Some statements in the literature review require references, but all information is relevant to this research. The structure of the paper is appropriate, and the figures are helpful and high-quality. The raw data is consistent with what is reported in the manuscript.

Specific issues and questions I have are:

1. L46-48: There should be a direct citation for the quantities listed in this statement.
2. L83-85: None of the references in this paragraph mention reliability issues due to image noise. In fact, Brandoli et al. include a processing step to remove noise. This statement needs further justification or should be removed.
3. L117 & Fig. 3: The word “massive” usually refers to a single large measurement, but the number of droplets is plural. “Massive” should be replaced with another adjective, such as “many”, to make it more clear that the authors are referring to the number of droplets.
4. L88-89: The genetic algorithm and the Otsu algorithm should be cited. It would also be helpful to include a general description of how they work, and some examples of their uses in the literature which justify why they may help extract droplets on WSPs.
5. L77: Özlüoymak et al. 2019 is not in the references list. Did the authors mean to cite Özlüoymak & Bolat 2020 here, or a different paper?
6. L81: The reference “Bruno et al.” should be “Brandoli et al.”. Bruno is the author’s given name, not surname.
7. L45, 77, 81, 83, 135: There should be a space between the end of the statement and the beginning of the reference.
8. L222: There is a typo at the end of this line, “Gene-Otsu” instead of Genetic-Otsu.

Experimental design

This paper fits in the scope of PeerJ; there are several sections of the journal that this paper would fit into. The research investigates the use of three digital image processing techniques for extracting droplet information from water-sensitive papers (WSPs). The methodology is sound, but requires some additional detail:

Specifically:

1. L142: What value did the authors use for Threshold, and how did they choose it? This value is required to replicate the experiment.
2. L197: Manual measurement of this many droplets could be prone to human error. Why did the authors use a manual method instead of a control spray card like the one used by Brandoli et al. (2021)?
3. L172: How does one know that the genetic algorithm found the optimal solution? Is there a ground truth to compare against?
4. L96: Are there any specific reasons the authors chose this drone over other commercially available options such as the DJI Agras T40?

Validity of the findings

The authors cited three other papers (Ferguson et al., 2016; Özlüoymak & Bolat 2020; Brandoli et al., 2021) which used digital image processing techniques to obtain droplet information from WSPs, and claimed that image noise caused reliability issues in the detection of small droplets. These reliability issues were used as a justification for this study (L83-88). The authors have not proven that their image processing methods improve reliability because they have not tested their images using the other cited image processing methods. If the authors analyze their WSP images using the software cited in their literature review, they would provide stronger support for their Genetic-OTSU method.

Additional comments

Overall, this paper has good scientific merit. With some revisions, it will be an excellent contribution to this journal and the scientific community. There are multiple methods of obtaining droplet information from WSPs, without a single defined standard. This paper provides another option for farmers, agricultural specialists, and scientists which may best fit their use case.

·

Basic reporting

Introduction, objective of study, justification is not clear. It may be modified.

Experimental design

Statistical research plan is not in manuscript. It is required to strengthen the result and study

Validity of the findings

Droplets deposition characteristics need to validate

Additional comments

1. Objective of study and justification are not clear. It may be modified. More reference may be considered for introduction and justification.
2. In table 1: what is Drone quality (kg), 12
3. Line 106: In fig. 2, which method is adopted to decide water sensitive paper distance 0.4 m and what is the nozzle overlapping?
4. Line 109 and 110: swath depends on height of drone, how 4 m of swath is fixed.
5. Experimental design section, statistical research plan may be clearly defined and added in tabular form in terms of independent parameters, dependent parameters and level and replication and droplets parameters consider for study.
6. What is the size of water sensitive paper has been considered for study.
7. droplet deposition parameters consider VMD, NMD, % coverage, droplets density, deposition, etc., This is missed in manuscript, It may be added otherwise title of manuscript is incomplete
8. Droplets deposition study must be done both surface of leaf adaxial and abaxial because insects are also hidden back side of leaf. So % coverage must be considered for both surfaces of leaf for insect kill efficacy.
9. Droplets analyzing process is not clear, please explain it.

---

## Round 0.2 · accepted · Accept

Your revised manuscript, which addresses the reviewers’ concerns and suggestions, is acceptable. Congratulations

·

Basic reporting

no comment

Experimental design

no comment

Validity of the findings

no comment

Additional comments

The authors did an excellent job addressing my concerns and suggestions from the previous review.